# Computational Pipeline for Reference-Free Comparative Analysis of RNA 3D Structures Applied to SARS-CoV-2 UTR Models

**DOI:** 10.3390/ijms23179630

**Published:** 2022-08-25

**Authors:** Julita Gumna, Maciej Antczak, Ryszard W. Adamiak, Janusz M. Bujnicki, Shi-Jie Chen, Feng Ding, Pritha Ghosh, Jun Li, Sunandan Mukherjee, Chandran Nithin, Katarzyna Pachulska-Wieczorek, Almudena Ponce-Salvatierra, Mariusz Popenda, Joanna Sarzynska, Tomasz Wirecki, Dong Zhang, Sicheng Zhang, Tomasz Zok, Eric Westhof, Zhichao Miao, Marta Szachniuk, Agnieszka Rybarczyk

**Affiliations:** 1Institute of Bioorganic Chemistry, Polish Academy of Sciences, 61-704 Poznan, Poland; 2Institute of Computing Science, Poznan University of Technology, 60-965 Poznan, Poland; 3Laboratory of Bioinformatics and Protein Engineering, International Institute of Molecular and Cell Biology in Warsaw, 02-109 Warsaw, Poland; 4Department of Physics, Department of Biochemistry, Institute for Data Science and Informatics, University of Missouri, Columbia, MO 65211, USA; 5Department of Physics and Astronomy, Clemson University, Clemson, SC 29634, USA; 6Laboratory of Computational Biology, Faculty of Chemistry, Biological and Chemical Research Centre, University of Warsaw, 02-089 Warsaw, Poland; 7Architecture et Réactivité de l’ARN, Université de Strasbourg, Institut de Biologie Moléculaire et Cellulaire du CNRS, 67084 Strasbourg, France; 8Translational Research Institute of Brain and Brain-Like Intelligence, Department of Anesthesiology, Shanghai Fourth People’s Hospital Affiliated to Tongji University School of Medicine, Shanghai 200081, China

**Keywords:** multi-model evaluation, RNA 3D structures, 3D structure prediction, reference-free analysis, SARS-CoV-2 genome, 5′-UTR, 3′-UTR

## Abstract

RNA is a unique biomolecule that is involved in a variety of fundamental biological functions, all of which depend solely on its structure and dynamics. Since the experimental determination of crystal RNA structures is laborious, computational 3D structure prediction methods are experiencing an ongoing and thriving development. Such methods can lead to many models; thus, it is necessary to build comparisons and extract common structural motifs for further medical or biological studies. Here, we introduce a computational pipeline dedicated to reference-free high-throughput comparative analysis of 3D RNA structures. We show its application in the RNA-Puzzles challenge, in which five participating groups attempted to predict the three-dimensional structures of 5${}^{\prime }$- and 3${}^{\prime }$-untranslated regions (UTRs) of the SARS-CoV-2 genome. We report the results of this puzzle and discuss the structural motifs obtained from the analysis. All simulated models and tools incorporated into the pipeline are open to scientific and academic use.

## 1. Introduction

### 1.1. Research Context

Ribonucleic acids, once considered passive carriers of genetic information, are currently known to play a central role in virtually all cellular processes. They serve as templates for protein translation, regulate gene expression on many levels, and govern most aspects of cell metabolism, thus being crucial regulators of physiological and disease processes [1,2,3]. To exert their functions, RNAs adopt specific three-dimensional structures resulting from secondary and tertiary interactions. The dynamic nature of RNAs means that many functionally important native states of these molecules exist, such as folding intermediates or elements containing flexible motifs. Therefore, identifying functional RNA structures is particularly difficult and requires combining experimental and computational approaches [4,5,6,7].

Although RNA prediction and modeling develops rapidly and many new strategies have been introduced so far [8,9,10,11,12,13,14], the determination of the 3D structure of RNA remains challenging [15]. Existing RNA tertiary structure prediction methods can produce a number of models for a single sequence. They are usually clustered by similarity and filtered to select the best solutions, but the final result is rarely consistent with the native conformation [10,16]. This problem occurs especially for larger molecules (>100 nts), which contain multi-branched loops, non-canonical base-base and base-backbone interactions, and long-range tertiary contacts [14,17,18,19]. Choosing a good model for designing, testing, confirming, or rejecting chemical and biological hypotheses is complicated due to the deficiency of experimentally solved structures for most RNA molecules and the randomness present in many 3D RNA modeling algorithms [18,20]. Researchers who would like to assess predicted models without the context of crystal structures as a reference, evaluate prediction methods, or explore similarities and differences between obtained predictions face a non-trivial task. Existing comparative approaches often impose limits on the size of molecules, preventing the processing of large samples [21,22,23].

Here, we present the analytical pipeline for reference-free high-throughput comparative analysis of RNA 3D structures. We show its first application in the RNA-Puzzles challenge that represented the community effort to predict the three-dimensional structures of functionally significant RNA elements in the SARS-CoV-2 genome, the 3${}^{\prime }$-UTR and 5${}^{\prime }$-UTR together with adjacent coding regions. The challenge was entered by five modeling groups, who submitted their predictions; one of these (the Das group) has published its predictions separately [24]. The Szachniuk group, responsible for the analysis of all generated models, designed and applied the presented pipeline. It comprises an RNA 3D structure study at the secondary and tertiary structure levels, including extraction and comparison of the RNA 2D and 3D domains. It is the first application of such a comprehensive approach in RNA structural bioinformatics. Due to the availability of computational tools used in the proposed pipeline, its scheme can be easily followed by other researchers.

### 1.2. Biological Background

Viral RNA genomes are the subject of extensive structural research. Currently, due to the COVID-19 pandemic, the most intensively studied one is that of severe acute respiratory syndrome coronavirus 2 (SARS-CoV-2). Several research groups provided secondary structure models of entire SARS-CoV-2 genome or its fragments determined in various experimental states [15,25,26,27,28,29,30,31]. These studies and computational predictions [32,33] have shown that the 5${}^{\prime }$ and 3${}^{\prime }$ UTR of the SARS-CoV-2 RNA genome adapt complex secondary structures.

The 5${}^{\prime }$ UTR of SARS-CoV-2 RNA (265 nt) together with cis-acting sequences that extend 3${}^{\prime }$ of the 5${}^{\prime }$-UTR into ORF1a (28 nt) comprises five stem-loop structures SL1–SL5. The first three hairpins—highly conserved across coronaviruses [34,35]—are small structural motifs formed by 4–10 base pairs and apical loops of several nucleotides. SL1 is crucial to relieve inhibition of mRNA translation induced by the NSP1 protein and to allow the production of SARS-CoV-2 proteins [36,37]. SL3 exposes the transcriptional regulatory core sequence (TRS-L: 5${}^{\prime }$-ACGAAC-3${}^{\prime }$), which is essential for discontinuous transcription [35]. It is also involved in the cyclization of SARS-CoV-2 gRNA [38]. The downstream 5${}^{\prime }$ UTR region consists of two hairpins, SL4 and SL5abc. The SL4 region is presented in two 2D model versions: a bipartite domain structure that includes two stem loops (SL4, SL4.5) or a hairpin and longer single-stranded region between SL4 and SL5. SL4 contains a short open reading frame (uORF) that, according to studies of other CoVs, serves as a negative regulator of downstream ORF translation [39]. SL5abc constitutes the largest structural motif in the 5${}^{\prime }$-UTR and includes the trifurcated stem formed by long-range base-pairing between 5${}^{\prime }$-UTR and the ORF1a. The three apical loops contain sequence motifs (UUCGU in SL5a/b and GNRA in SL5c) characteristic of other coronaviruses and probably involved in the packaging of RNA in virions [40]. The region between SL4 and SL5 has been shown to be the preferred binding site for the SARS-CoV-2 nucleocapsid protein (N), likely significant for packaging [41].

The 3${}^{\prime }$ UTR of the SARS-CoV-2 genome (337 nts) contains the bulged stem loop (BSL) and the P2 motif. For other CoVs, it has been shown that BSL can form a pseudoknot (PK) with the neighboring hairpin [42,43,44], but for SARS-CoV-2, so far, no evidence supports the formation of such a structure. The downstream region of the P2—that is, the highly variable region (HVR)—forms a long-bulged stem and contains the highly conserved stem loop II motif (S2M) with a GNRA-like pentaloop. It is hypothesized that S2M is involved in intermolecular RNA/RNA interactions and the formation of a kissing loop dimer [45]. Furthermore, HVR contains the octanucleotide sequence (5${}^{\prime }$-GGAAGAGC-3${}^{\prime }$), characteristic for all CoVs, assumed to have a critical biological function [46].

5${}^{\prime }$ and 3${}^{\prime }$ UTR as conserved regulatory RNA motifs in the SARS-CoV-2 genome are attractive potential targets for small-molecule drugs, but little is known about their 3D shapes. Several research groups presented in silico generated 3D models of individual UTR motifs [24,25,47,48]. However, the 3D structures of all SARS-CoV-2 UTRs still need a thorough investigation.

## 2. Results

Here, we demonstrate the use of our computational pipeline for the SARS-CoV-2 challenge of RNA-Puzzles. The participants in this contest submitted 74 3D models of RNA, of which 34 concerned 5${}^{\prime }$-UTR, and 40 referred to 3${}^{\prime }$-UTR (cf. Table 1).

For 5${}^{\prime }$ UTR, the modeling groups decided to submit 3D RNA structures generated for different lengths ranging from 268 to 450 nucleotides. Since structural and genetic studies indicate that cis-acting sequences that extend 3${}^{\prime }$ of the 5${}^{\prime }$-UTR into ORF1a play an essential role in viral RNA synthesis and fold into a set of highly ordered and well-conserved secondary structure elements of RNA (i.e., domains, stem loops) [24,25,27,49], the 268 nt 5${}^{\prime }$-UTR region is usually extended by 25 residues that are an integral part of the fifth stem loop (SL5 domain). Therefore, we rejected 3D models of size smaller than 293 nt, and the analysis was performed in the common length variant of 293 nt (see Table 1).

The results obtained are described in more detail in the following.

### 2.1. Analysis of SARS-CoV-2 5${}^{\prime }$-UTR Models

The analysis includes 34 structures predicted by four research groups (Table 1).

#### 2.1.1. 3D Structure Data Normalization and Validation

We have evaluated the stereochemical accuracy of the 3D structures submitted, and, based on the results obtained, we concluded that they are consistent with those presented in the summary of round IV of RNA-Puzzles [15]. Models from two groups, Das and Szachniuk, have significantly fewer stereochemical inaccuracies compared to the other submissions (cf. Appendix A).

Using the RNAspider pipeline [7], we found and classified entanglements of structure elements, which turned out to be present within two of 34 RNA 3D models provided by the Bujnicki group (see Appendix A).

#### 2.1.2. Global RMSD-Based Pairwise Comparison of RNA 3D Models

In the next step, the global pairwise comparison of all 3D models was conducted. We could observe that, in general, the submitted models were diverse in their global 3D folds. However, significant similarities can be detected between the 3D RNA structures submitted by a given group (cf. Appendix A illustrated by a colored heat map based on RMSD scores). This effect comes from the strategies adopted by different modeling teams. In other words, some predictors generated large 3D RNA structure ensembles, clustered them, and submitted the top-scoring cluster members, which diversified the overall collection, while other groups did not follow this approach, allowing for similar models within the submission. Therefore, we were particularly interested in the comparison of 3D models from different modeling teams.

For the ensemble of models, we calculated the values of the extreme and average RMSD values together with standard deviation, and determined the member of the top scoring ensemble (the centroid of the whole ensemble) with the average distance to it (cf. Table 2 and Figure 1).

Based on the obtained results (cf. Appendix A and Table 2 and Figure 1), it can be noted, that the majority of similar 3D structures were observed between Das and Szachniuk groups, with minimum RMSD equal to 25.411 Å. Here, six pairs of models were assigned RMSD scores below 3 Å and 5UTR-Das-01_1 model was essentially similar to four Szachniuk group models (RMSD lower than 30 Å).

The major differences between models submitted by different groups could be detected for Bujnicki and Szachniuk teams, with an RMSD score equal to 64.384 Å. The RNAComposer-5UTRD_3 model was the most diverged one as compared to the ensembles of 3D structures submitted by different predictors (RMSD average value equal to 47.14 Å).

#### 2.1.3. RNA 2D Structure

First, we extracted secondary structures from 3D structure atom coordinates, and we prepared conservation logos (cf. Figure 2). Next, a pairwise comparison of all considered secondary structures was performed. As a result, six clusters were obtained, of which five consisted mainly of RNA 2D structures derived from models submitted within a given group (cf. Appendix A) and one was composed of submissions originated from the Das and Szachniuk groups. This suggests that the submitted models tend to be diverse in their secondary structure.

In the next step, a consensus-driven approach was applied to identify RNA domains within submitted models. Consequently, seven conserved elements were identified (cf. Table 3 and Figure 3).

Next, utilizing the domain-boundary-driven approach, we obtained 28 groups of domains, where 13 of them contained segments derived from models submitted by at least two different predictors (cf. Appendix A). Moreover, 7 out of 13 were present in the vast majority of all 3D RNA models (represented in red in Appendix A). According to the published data [24], they corresponded to the following domains: SL1 (7–33 nt), SL2 (45–59 nt), SL3 (61–75 nt), SL4 (84–127 nt), SL5a (188–218 nt), SL5b (228–252 nt) and SL5 stem (151–182 nt, 263–293 nt) (cf. Figure 3).

#### 2.1.4. RMSD-Based Pairwise Comparison and Clustering of RNA 3D Domains

The results of this step are presented and summarised in Table 3. Note that all high-order and highly conserved domains reported in recent literature [24,25,26,27,28,29,30,31,49] have also been found in the 3D RNA models considered in this study (cf. Figure 3).

Next, we conducted a clustering-based analysis of coaxial helical stacking for SL5abc four-way junction (4WJ) and two domains, namely SL2 and SL3 (since it contains an important transcription-regulating (TRS-L) sequence required for subgenomic viral RNA synthesis [51]) within the 5${}^{\prime }$-UTR region.

An SL5abc four-way junction occurred in 24 of 34 models. In each case, 4WJ had no single-stranded region between consecutive helices (cf. Figure 3).

As a result, we could observe that most of the models were characterized by two coaxial helices. Members of the largest cluster (nine models submitted by the Szachniuk group) belonged to the family cH [52] with two pairs of coaxial stacks SL5-stem/SL5a and SL5b/SL5c (cf. Figure 4), while the other cluster of family cH (five members) displayed a coaxial stacking pattern SL5-stem/SL5c and SL5a/SL5b. Two other clusters (five and four members, respectively) represented family H.

Finally, the analysis of Figure 4A showed that models belonging to a given family of 4WJ still displayed a wide variability because of a large asymmetric internal loop in the SL5a_ext region that could cause a kink and therefore a different spatial arrangement of SL5a.

Next, we conducted the analysis of the mutual arrangement of the SL2 and SL3 domains. Both of them appear together in 32 out of 34 such models. The models are characterized by a large variation in the mutual arrangement of SL2 and SL3. Most of them have a bend at the unpaired U60 connecting the SL2 and SL3. Only a few models have roughly coaxial stacking of SL2 and SL3 stems (two models from the Das group, cf. Figure 5). In 24 of the 32 models, U30 is in the stacking interactions with the bases closing at least one stem. For 2 of the 32 models, U60 stacks both with SL2 and SL3, while in 16 models, U60 stacks only with SL2, and in 6 models, U60 stacks only with SL3. In 8 of the 32 models, U60 has no stacking interactions with any of the SL2 or SL3 stems.

#### 2.1.5. The Comparison of Predicted SL2 Domain with Reference Experimental NMR 2L6I Structure (Solution Structure of Coronaviral Stem-Loop 2)

The structure of SL2 domain has previously been determined for SARS-CoV-1 using NMR spectroscopy [53]. This domain is conserved among SARS-CoV-2 and SARS-CoV-1. The comparison between SL2 hairpin models and the NMR structure (model 1) showed that almost 32% of all 3D structures (11/34) have RMSD below 2 Å, whereas four models provided by then Chen group have an RMSD less than 1 Å. The detailed results are shown in Appendix A. The NMR structure has an additional C50-G53 base pair within the loop. This base pair is present in 11 models (out of 35), constituting all of Chen’s models (1–10) and one of Bujnicki’s model (Bujnicki_04). Moreover, within three models of the Bujnicki group (02, 03 and 05), this loop is involved in high-order interactions.

### 2.2. Analysis of SARS-CoV-2 3${}^{\prime }$-UTR Models

The analysis included 40 structures predicted by five research groups (Table 1).

#### 2.2.1. 3D Structure Data Normalization and Validation

Similar to the case of the 5${}^{\prime }$-UTR analyses, we evaluated the stereochemical accuracy of the submitted 3D structures, and we concluded that they are consistent with those presented in the RNA-Puzzles round IV summary [15]. Note that within the models from the Das and Szachniuk groups, considerably fewer stereochemical inaccuracies were identified as compared to those submitted by other groups (cf. Appendix A).

Using the RNAspider [7], we found and classified entanglements of the structural elements, which appeared in 4 non-pseudo knotted 3D RNAs and in 15 pseudoknotted models (cf. Appendix A). This is consistent with previous results where it was shown that entanglements of structural elements tend to appear in RNA 3D structures with higher-order interactions [7].

#### 2.2.2. Global RMSD-Based Pairwise Comparison of RNA 3D Models

The global pairwise comparison of all 3D models was conducted similarly to that of the 5${}^{\prime }$-UTR 3D RNA structures. We could observe that in general, they are very diverse (cf. Appendix A illustrated by a coloured heat-map based on RMSD scores). As in the case of the 5${}^{\prime }$-UTR, similar trends can be observed as the outcome of the strategies adopted by the different predictors.

Among the submitted 3D structures, the Chen and Das groups modeled a putative pseudoknotted conformation with base pairs between BSL and P2 domain, whereas other models represented the 3${}^{\prime }$-UTR as a non-pseudoknotted structure. Although the presence of a pseudoknot in 3${}^{\prime }$-UTR of SARS-CoV-2 RNA is not supported by the recent experimental data [26,28,29,31,38], it was shown to be conserved in beta- and alphacoronaviruses [54,55]. Therefore, we decided to divide the submitted 3D RNA structures into two sets, those composed of pseudoknotted and non-pseudoknotted structures, and analyse them separately. For each ensemble of models, we calculated extreme and average RMSD values together with standard deviation, and we determined the top scoring ensemble member (the centroid of the whole ensemble) with the average distance to it (cf. Table 2, Figure 6 for non-pseudoknotted structures and Appendix A for pseudoknotted structures).

Based on the obtained results (cf. Appendix A, Table 2 and Figure 6), it can be observed that the average RMSD between a given 3D structure and the structures from other modeling groups is in the range of 40–50 Å, with the exception of 5 models, where it exceeds 50 Å. The 3UTR-Das-04_1 model is, on average, the most similar to the other models, apart from Das’ group (average RMSD is equal to 41.285 Å). The most similar 3D structures across the groups are 3UTR-Bujnicki-02 and 3UTR-Chen-1_1 (24.907 Å), and the most different ones are 3UTR-Bujnicki-03 and 3UTR-Ding_2 (84.287 Å).

#### 2.2.3. RNA 2D Structure

In this step, conservation analysis was carried out based on secondary structures extracted from 3D structure atom coordinates. As a result, a conservation logo was calculated (Figure 7), which shows that although there are evident differences between the analysed models, some regions tend to be well conserved. To further investigate these similarities, we conducted an analysis of the shorter elements within the considered structures (domains).

Next, pairwise comparison of all secondary structures obtained in the previous step was performed. As a result, six clusters were obtained, all of which consisted only of RNA 2D structures derived from models submitted by single modeling groups (cf. Appendix A). This indicates that from a global perspective, all the submitted models tend to be diverse (cf. Figure 7).

In the following step, we applied a consensus-driven approach for domain identification to find the longest possible elements that are closed by base pairs and that are common to at least 50% of considered models. Based on the results of this analysis, we identified four such elements (cf. Figure 8 and Table 4).

Next, using a domain-boundary-driven approach, we extracted all possible domains, even when they were present in less than 50% of the models (see Section 4). All the identified domains were then grouped by sequence. As a result, 52 groups of domains were obtained, 14 of them containing segments derived from models submitted by at least two different modeling groups (cf. Appendix A).

Domains extracted using above analyses were in agreement with published consensus structure [24] and corresponded to the following domains: BSL (26–72 nt), P2 (96–124 nt), HVR (172–186 nt), HVR-stem (128–170, 268–317 nt) (cf. Figure 8).

#### 2.2.4. RMSD-Based Pairwise Comparison and Clustering of RNA 3D Domains

The results of this step are presented and summarised in Table 4.

From these analyses, we concluded that the most conserved domains within the 3${}^{\prime }$-UTR region were the following: the BSL, P2, HVR-hairpin, and the HVR-stem. Although elements such as BSL-ext and S2M were less conserved in comparison to the former, they were still preserved in most of the submitted 3D RNA models.

Additionally, we conducted a clustering-based analysis of coaxial helical stacking for the HVR hairpin and HVR stem domains. The HVR hairpin domain occurred in 35 out of 40 models, of which 20 models represented HVR hairpin in coaxial arrangement with the HVR stem (cf. Figure 9).

#### 2.2.5. The Comparison of Predicted S2M Domain with Reference Experimental Crystal G225U Structure from SARS-CoV-1

The three-dimensional crystal structure of S2M has been solved for the SARS-CoV-1 virus genome [56]; thus, we conducted the comparison between the S2M from submitted models of the 3${}^{\prime }$-UTR and the reference X-ray structure (G225U in SARS-CoV-1). As a result, we observed a very similar S2M structure for the Szachniuk group models (RMSD in the range of 1.82–2.24 Å), while the models submitted by other groups contained more diverse and different S2M predictions (RMSD ranging between 6.85 Å and 13.25 Å). The detailed results are shown in Appendix A.

## 3. Discussion

The knowledge of detailed 3D structures is essential to recognizing and understanding the biological function of RNA molecules and for drug design. Unfortunately, there is still a noticeable gap between known RNA sequences and the number of experimentally determined structures, which highlights the importance and further stimulates the development of computational 3D structure prediction methods. Consequently, it motivates researchers to compare models and select the most representative 3D structural motifs based on the set of 3D structures generated, especially in the absence of the reference structure [13,20]. The analytical pipeline for the reference-free high-throughput comparative analysis of 3D structures of RNA described in this paper addresses these expectations.

Furthermore, with the SARS-CoV-2 pandemic outbreak in 2019, unprecedented support and collaboration among scientists has emerged, resulting in noticeable advances in the field of structural and computational biology, thus triggering efforts towards a comprehensive structural analysis of the SARS-CoV-2 genome to better respond to this deadly disease. First, 2D structure models of the partial or complete SARS-CoV-2 genome were determined in various experimental states (in vitro, in virio, in vivo, and ex vivo, extracted and followed by refolding in vitro) [15,25,26,27,28,29,30,31]. Unfortunately, beyond the secondary structure of the conserved regions of the SARS-CoV-2 RNA genome, little is still known about their 3D structural representation. A recent work presented de novo modeled 3D structures of individual motifs from the UTRs and a 3D model of the FSE [24]. Furthermore, 3D models of highly structured regions of the SARS-CoV-2 genome and proposed potential ligand binding pockets in 3D structures of RNA are available [25,47,48]. However, the 3D structures of all SARS-CoV-2 UTRs need to be thoroughly studied and investigated. In-depth knowledge of the 3D structure of these highly conserved regulatory RNA elements is the key to advancing the development of novel antiviral therapies.

Therefore, the RNA-Puzzles community decided to contribute to these efforts and announced an additional prediction challenge regarding the determination of 3D structures of functionally important RNA elements in the SARS-CoV-2 genome, namely the 3${}^{\prime }$-UTR and 5${}^{\prime }$-UTR, together with adjacent coding regions. Since each modeling group obtained its 3D structures using various prediction strategies, the evaluation and comparison of such a heterogeneous and large set of models represented a challenge. In order to tackle this problem, we have applied the computational pipeline designed by the Szachniuk group and described in this paper. To our knowledge, this is the first such extensive and holistic approach in RNA structural bioinformatics.

As a result, it turned out that an aforementioned pipeline is especially useful in the case of complex, multi-domain models, where the RMSD-based global pairwise comparison of 3D structures is not informative since the global folds of molecules are too diverse, and thus it is necessary to focus on the 3D structure of single, key regulatory structural regions (domains).

To characterize and identify common structural motifs in the generated 3D models of SARS-CoV-2 RNA, we computed consensus secondary structures of 5${}^{\prime }$-UTR and 3${}^{\prime }$-UTR. The generated consensus models of the 5${}^{\prime }$-UTR are generally in good agreement with the experimentally confirmed structures obtained by SHAPE or DMS mapping of the whole SARS-CoV-2 RNA genome [25,28,29,31]. Most models contain conserved stem loop motifs SL1, SL2, SL3 and SL4 among various CoVs. Almost all models have SL5a, SL5b and SL5c that are connected to a four-way junction. In all models, the SL1 has a 5${}^{\prime }$-UCCC-3${}^{\prime }$ apical loop and long bipartite stem interrupted by a 3-nt internal loop or a single nucleotide bulge and non-canonical base pair. In other CoVs, SL1 is structurally and functionally bipartite, as mutations that alter the pairing of the bases in the upper and lower SL1 stem differentially affect virus replication [57]. Analysis of emerging variations within the cis-regulatory RNA structures of the SARS-CoV-2 genome showed that SL1 is a hot spot for viral mutations. Interestingly, most of them stabilize the structure of SL1 by increasing the length of their stem [58], which may suggest that stabilization of SL1 does not have deleterious effects and may even be significant in SARS-CoV-2 replication.

Next, all structures contain a similar SL2 motif with a conserved pentaloop that has been shown to be critical for subgenomic RNA synthesis [34]. In some models, the apical loop of the SL2 is stabilized by a cross-loop G-C base pair. SL3 with the TRS-L sequence located in the apical loop and 3${}^{\prime }$ stem (nt 70-75) is present in almost all models predicted for 5${}^{\prime }$-UTR sequence. Due to the high content of the base coating A-U, the stem of SL3 is relatively thermodynamically unstable, and recent studies have shown that the SL3 sequence can be involved in the cyclization of genomic SARS-CoV-2 RNA mediated by a long-range interaction between the +60–80 region in 5${}^{\prime }$-UTR and +313–334 in 3${}^{\prime }$-UTR [38]. The mentioned 3${}^{\prime }$-UTR region in our models is also partially single-stranded, which may indicate the formation of such an interaction. In particular, since hairpin SL3 contains the TRS-L sequence, it is possible that genome cyclization regulates sgRNA synthesis. During discontinuous transcription, a replication and transcription complex (RTC) starts RNA synthesis from the gRNA 3${}^{\prime }$ end, pauses on specific sites containing transcription regulatory sequence (TRS-B) located upstream of each ORF, and changes the template, probably through another RNA-RNA interaction between TRS-L and TRS-B, bypassing the internal regions of the gRNA [26].

In most models, SL4 adopts a bipartite domain structure that includes two stem-loop motifs, SL4 and SL4.5. The start codon of conserved uORF is found in the loop of SL4, while the 3${}^{\prime }$ part of uORF is in the stem of SL4.5. A bipartite structure of the SL4 motif was also proposed, with shorter SL4.5, for the 2D model of SARS-CoV-2 genomic RNA in vivo by the Pyle group [28]. The other experimentally determined models of the SARS-CoV-2 genome contain a shorter SL4 motif and a single-stranded conformation of the 3${}^{\prime }$ part of uORF that is more similar to that proposed for MHV, BCoV, and SARS-CoV [35]. A single form of the SL4 motif is also found in some models, but the uORF sequence is base-paired and forms an elongated stem of SL4.

The SL5 motif has common features in most models, including 5${}^{\prime }$-UUUCGU-3${}^{\prime }$ apical loops on SL5a and SL5b and a 5${}^{\prime }$-GNRA-3${}^{\prime }$ tetraloop on SL5c, which are believed to act as a packaging signal. The difference can be seen in the Chen group model, where SL5b is longer and the SL5c motif is missing. Interestingly, the 2D structure generated for models of the Bujnicki group contains pseudoknot motifs that are formed between the SL2 loop and the single-stranded region downstream of SL4, and the SL3 loop and the single-stranded region downstream of SL1. Recently, the presence of pseudoknots in the 5${}^{\prime }$-UTR was also proposed based on the in vitro mapping of the SARS-CoV-2 structure, but they engage different nucleotide sequences [15].

For the 3${}^{\prime }$ terminus, predictions were made for the region of 1–337 nt. All consensus 2D structures contain a BSL motif, but with different stem lengths and amounts and positions of mismatches and bulges. All 2D models also contain P2 with a large, 11-nt apical loop. Chen and Das groups proposed a pseudoknot formed between a single-stranded region downstream of BSL and the apical loop of P2. However, models from other groups present the 3${}^{\prime }$-UTR as a non-pseudonotted structure. Although the presence of pseudoknot in 3${}^{\prime }$-UTR was predicted to be conserved in beta and alphacoronaviruses [54,55], recent experimental data do not support the folding of the stem-loop pseudoknot in the 3${}^{\prime }$-UTR of SARS-CoV-2 RNA in vivo [26,28,29,31,38].

The hypervariable region (HVR) containing the long-bulged stem covers almost the same range of nucleotides in all consensus structures, but differences can be observed in the number of mismatches and the location and size of the bulges. The HVR is defined as structurally dynamic; therefore, a different model is not surprising. The presence of multiple mutations in this region of 3${}^{\prime }$-UTR was shown for SARS-CoV-2, suggesting that the HVR is not important for its replication [58]. HVR is poorly conserved in CoVs, and mutational tests in MHV showed that a significant part of this region is not essential for viral RNA synthesis [46]. However, it contains the conserved octa-nucleotide motif 5${}^{\prime }$-GGAAGAGC-3${}^{\prime }$, which is assumed to have a critical biological function [46]. This motif is situated between nucleotides 261–268 and in most models appears in a single-stranded conformation, which can facilitate protein binding.

The consensus models of 3${}^{\prime }$-UTR also include the S2M subdomain with a GNRA-like penta-loop and topology consistent with the crystal structure of S2M solved for SARS-CoV-1 [56]. A structure similar to S2M was observed for the 2D models that the Ding, Das and Szachniuk groups analyzed independently. The models for the Chen and Bujnicki groups contain different and unique S2M structures.

The 3D models of conserved structural RNA elements (domains) identified as a result of the comparative reference-free analysis presented in this paper can be used n ext, e.g., as new molecular targets to find drugs or in simulation experiments to investigate various biological hypothesis. For instance, SARS-CoV-2 5${}^{\prime }$-UTR structures obtained from modeling were used for virtual docking simulations of amiloride-based small molecules [59]. The RNAComposer model of the 5${}^{\prime }$-UTR stem loop SL1 was used to investigate its binding to the nonstructural protein 1 of SARS-CoV-2 (nsp1) using MD simulation [60]. Furthermore, the homology model for the SARS-CoV-2 stem loop II motif (S2M) was explored as a potential drug target by docking a library of FDA-approved drugs [61].

## 4. Materials and Methods

### 4.1. A Choice of Input RNA Sequences

The presented pipeline was applied to analyze the 3D structures predicted from the first complete sequence of SARS-CoV-2 (MN908947.3) reported in [28,31]. Input sequences of 293-nucleotide 5${}^{\prime }$-UTR and 337-nucleotide 3${}^{\prime }$-UTR are given in the Appendix A.

### 4.2. 3D Structure Prediction Methods

The modeling groups participating in the challenge applied different prediction methodologies and protocols. They are briefly described in the Appendix A for four groups. The fifth group has presented its approach in a separate paper [24].

### 4.3. Methods for RNA 3D Structure Analysis

The proposed pipeline for reference-free comparative analysis of 3D RNA structures consists of seven fundamental steps that are performed sequentially (Figure 10).

Step 1.Normalization and validation of 3D structure data.RNA 3D structure evaluation was conducted using rna-tools [62,63]. RCSB MAXIT was applied to evaluate the stereochemistry of the 3D structures submitted [64,65]. The RNAspider pipeline [7,66] was applied to identify and classify entanglements of structural elements, that is, spatial arrangements involving two structural elements, where at least one punctures the other. In this context, puncture refers to the situation in which one structural element (determined by the secondary structure of the molecule) intersects the area within the other [7].Step 2.RMSD-based global pairwise comparison of 3D RNA models. We compared the predicted models based on the root-mean-square deviation (RMSD) [67]. It was computed with the RNA QUality Assessment tool (RNAQUA) [62] for every pair of models. RMSD-based heat maps were prepared to support identification of similarities in the set of predictions. OC cluster analysis program was run with the default settings (single linkage algorithm) to calculate the centroids of the 3D structure ensembles of the RNA [68].Step 3.RNA 2D structure extraction from atom coordinate data and conservation analysis.RNApdbee [69,70] was applied to extract and annotate secondary structures from the atomic coordinates of all models. Based on multiple alignments of secondary structures, conservation logos were prepared using the WebLogo integration script [71].Step 4.Determination of the 2D consensus structure.The RNAtive tool [72] together with the consensus-driven approach for the identification of domains based on the RNA secondary structure (see the identification and analysis of domains of the RNA secondary structure for more details) was used to identify a consensus on all secondary structures annotated from the 3D input models of the RNA. RNAtive was performed with the predefined confidence threshold value set to 0.51. First, the interaction network for each input 3D model of RNA was calculated. Next, a consensus-driven secondary structure was calculated taking into account all interactions for which the confidence was higher or equal to the predefined threshold.Step 5.Clustering of the 2D structure of RNA.In this step, all secondary structures considered were compared pairwise using RNAdistance [73]. As RNAdistance does not handle pseudoknots, pseudoknot-forming nucleotides were treated as unpaired bases. Based on the comparison matrix obtained, the secondary structures were clustered with DBSCAN (density-based spatial clustering of noise applications) [74], a tool for data science and machine learning. It can identify clusters of varying shapes based on a user-defined distance measure and the minimum number of points required to find in proximity to create a cluster. Dimensionality was reduced using PCA (principal components analysis) [75].Step 6.The 2D structure-based identification of RNA domains and their analysis.A two-step approach was applied to identify RNA structure domains. In the first step (driven by consensus), the secondary structures of all RNAs considered were aligned. Next, the nucleotide pairing statistics were calculated, and the consensus secondary structure was generated and encoded in the extended dot-bracket notation. The consensus obtained was divided into domains. Each continuous fragment closed by base pairs, appearing in ≥50% of the models considered, was recognized as a domain. The aim was to find the longest possible elements closed by the base pairs common to ≥50% models. In the second step (domain-boundary driven), each RNA secondary structure was recursively divided into continuous domains. We aimed to enlarge the domains previously identified and make them more accurate. This approach resulted in a larger number of domains; some of them were present in ≤50% of models, while some were overlapping or were part of the larger ones. The base pairs involved in pseudoknot formation were independently analyzed as both unpaired and paired. With pseudoknots considered, a domain was defined as a continuous fragment located between corresponding structural elements that included opening and closing pseudoknot brackets. Such a routine was performed recurrently to enable the handling of small domains nested in the larger ones. Then, a statistical analysis of the identified domains was performed. Color-scaled maps of the regions analyzed were prepared, where the localization of the domains (Y-axis) was presented within the input sequence (X-axis). To perform a detailed analysis of the results, each domain was described by residue range, sequence, secondary structure, number of residues, number of participants who submitted models supporting the domain, distribution of the number of models within modeling groups, total number of models in which the domain was identified, and list of model names. Finally, all identified domains were aligned by an RNA sequence to observe the distribution of their secondary structures over the models.Step 7.RMSD-based pairwise comparison and clustering of RNA 3D domains.All domains identified in the previous step, supported by at least three different 3D models, were selected for further analysis. For each of them, the corresponding 3D substructures were extracted from all 3D models in which the domain was identified. For each 3D substructure, a pairwise comparison of the RMSD scores was performed, and an RMSD score matrix was prepared in a color scale. Furthermore, for each RMSD matrix, the mean and standard deviations were computed. Finally, for each domain independently, clustering was performed using DBSCAN [74] with a distance parameter set to 10Å based on the RMSD matrices. For each cluster of 3D RNA substructures, we computed extreme and average RMSD together with standard deviation, the highest scoring cluster member (the centroid of the cluster), the average distance to it, and the number of models within which a given domain was present.

### 4.4. Computational Platform and Parameters Used to Test the Pipeline

Computational experiments presented were run on a mobile workstation equipped with Intel Core i7-10850H CPU @ 2.70 GHz, RAM 64.0 GB, Ubuntu 18.04 LTS 64-bit. The consensus-driven evaluation was performed using RNAtive with all base pairs considered and a confidence level equal to 0.5. DBScan was executed with the minimal number of samples in a cluster equal to 2. The maximum distance between two samples for one to be counted as in the neighborhood of the other was 40 and max(10.0, 1.15*AVG RMSD) to cluster nonpseudoknotted secondary structures and domain RMSD scores. Scripts to perform the DBscan clustering and create secondary structure logos were developed in Python using the scikit-learn and weblogo libraries. Scripts to transform the data and process the results were developed in Bash and AWK.

## 5. Conclusions

In this study, we presented the computational pipeline for a reference-free high-throughput comparative analysis of the ensemble of 3D RNA structures. To our knowledge, it is the first such extensive and holistic approach in RNA structural bioinformatics.

We have demonstrated the first application of the pipeline in the RNA-Puzzles challenge and proved its contribution to understanding the structure of the SARS-CoV-2 virus and possible drug targets. Our analysis has shown that we are far from proposing reliable models for the entire UTR regions; however, individual domains can be modeled with high confidence, as shown by the consistency of 3D models for these domains obtained with different methods. Therefore, we have focused on the analysis of the three-dimensional structures of functionally important RNA elements (domains) in the SARS-CoV-2 genome.

Or study is the first one in which 3D RNA models of SARS-CoV-2 UTR regions generated by different modeling groups were evaluated and compared. The resultant 2D RNA consensus structures generated for the submitted 3D RNA models for both the 5${}^{\prime }$-UTR and 3${}^{\prime }$-UTR regions appeared to be in line with experimentally confirmed structures obtained by SHAPE or DMS [25,28,29,31]. All highly ordered and conserved domains within those regions reported in the work of [24,25,27,49] were also preserved in most 3D RNA models considered in this study.

The presented pipeline has the potential for various applications. It can be useful in comparative analysis of homologs aimed at identifying templates for homology modeling. It can also be applied to classify three-dimensional structures leading to reliable function prediction of biological molecules.

## Figures and Tables

**Figure 1 ijms-23-09630-f001:**
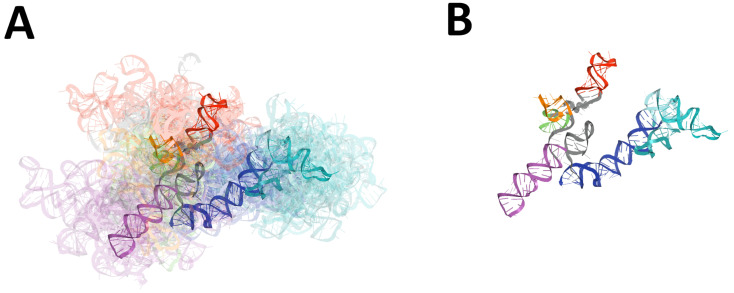
Results of global RMSD-based comparison of RNA 3D models predicted for the 5${}^{\prime }$-UTR region of SARS-CoV-2. Domains: SL1 (red), SL2 (green), SL3 (orange), SL4 (magenta), SL4.5 (light purple), SL5abc (cyan and teal), SL5 stem (blue). The centroid of each ensemble is solid, other members are transparent. (**A**) The ensemble of 3D RNA structures, and (**B**) its centroid.

**Figure 2 ijms-23-09630-f002:**
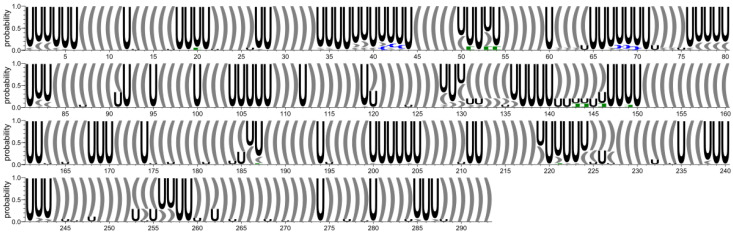
Secondary structure conservation diagram for the 5${}^{\prime }$-UTR region. ‘U’ corresponds to unpaired residue. According to the DBL representation of the secondary structure topology [50], square brackets [ ] (green) represent pseudoknots of the first order, while curly brackets { } (blue) represent second order pseudoknots.

**Figure 3 ijms-23-09630-f003:**
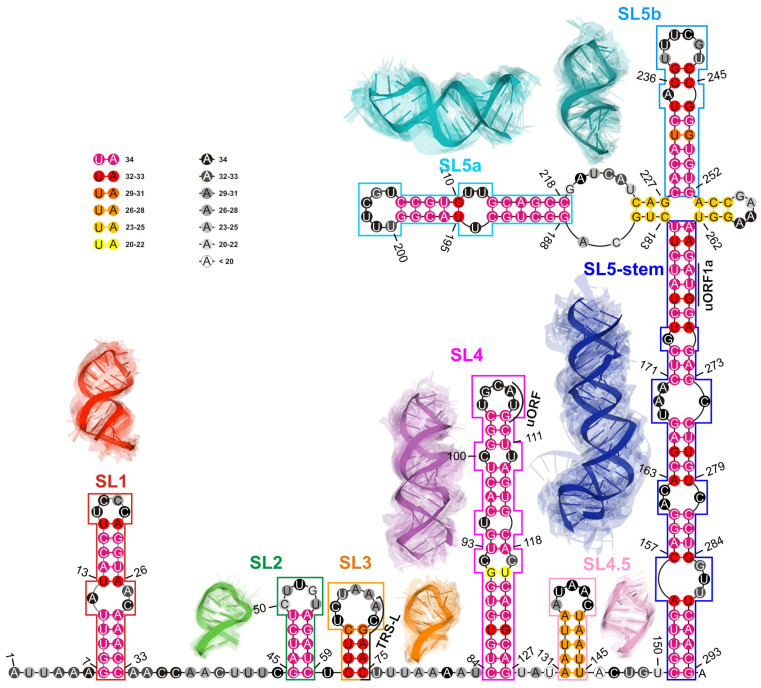
The consensus-driven secondary structure for the 5${}^{\prime }$-UTR region. Domains: SL1 (red), SL2 (green), SL3 (orange), SL4 (magenta), SL4.5 (light purple), SL5 stem (blue), SL5a (cyan), and SL5b (deep teal). Paired regions are coloured according to the preservation of base pairs in all considered 3D RNA models, from magenta (paired in 100% of models) to yellow (paired in ≥50% of models). Unpaired regions are coloured according to the probability that a given residue is not paired in all analysed models, from black (unpaired in 100% of models) to white (unpaired in ≥50% of models). Regions are bordered according to their colouring in 3D models. The centroid of each cluster is solid; the remaining cluster members are transparent.

**Figure 4 ijms-23-09630-f004:**
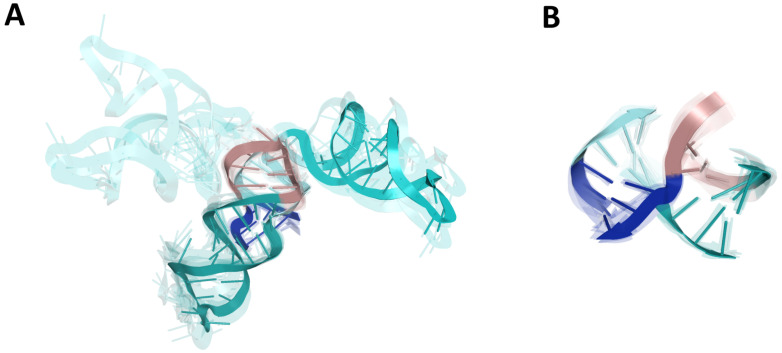
Four-way junction (4WJ) of SL5abc in the 5${}^{\prime }$-UTR region. (**A**) The ensemble of 3D RNA structures that belong to the cH family [52] with two pairs of coaxial stacks SL5-stem/SL5a and SL5b/SL5c that constitute the largest cluster (nine members). (**B**) 4WJ rotated 90-degrees around the y-axis. Domains: SL5a (cyan), SL5b (deep teal), SL5c (dirtyviolet), and SL5 stem (blue).

**Figure 5 ijms-23-09630-f005:**
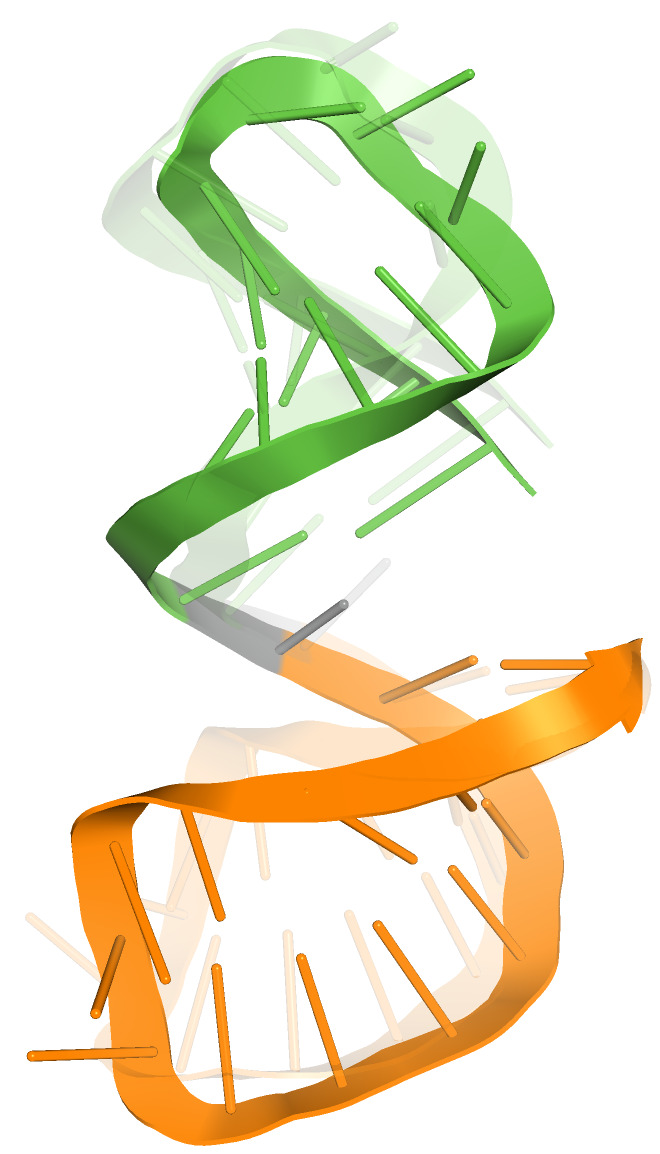
SL2 (green) and SL3 domain (orange) in roughly coaxial arrangement (cluster with two members).

**Figure 6 ijms-23-09630-f006:**
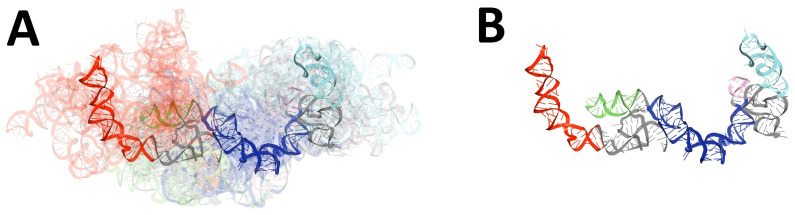
Results of global RMSD-based comparison of RNA 3D models predicted for 3${}^{\prime }$-UTR regions of SARS-CoV-2. Domains: BSL (red), P2 (green), HVR hairpin (light purple), SLM (cyan), HVR stem (blue). The centroid of each ensemble is solid, and the remaining members are transparent. (**A**) The ensemble of non-pseudoknotted 3D RNA structures and (**B**) its centroid.

**Figure 7 ijms-23-09630-f007:**
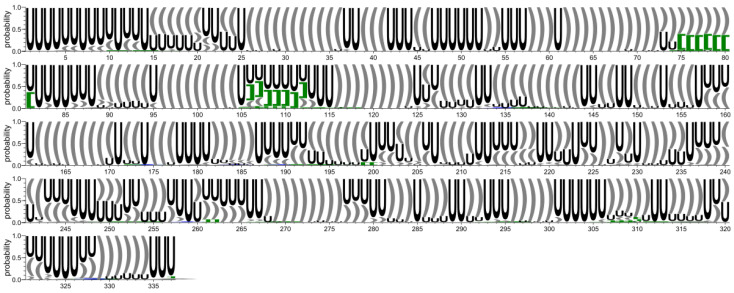
Secondary structure conservation diagram for the 3${}^{\prime }$-UTR region. ‘U’ corresponds to unpaired residue. According to the DBL representation of the secondary structure topology [50], square brackets [ ] (green) represent pseudoknots of the first order, and curly brackets { } (blue) represent the second order pseudoknots.

**Figure 8 ijms-23-09630-f008:**
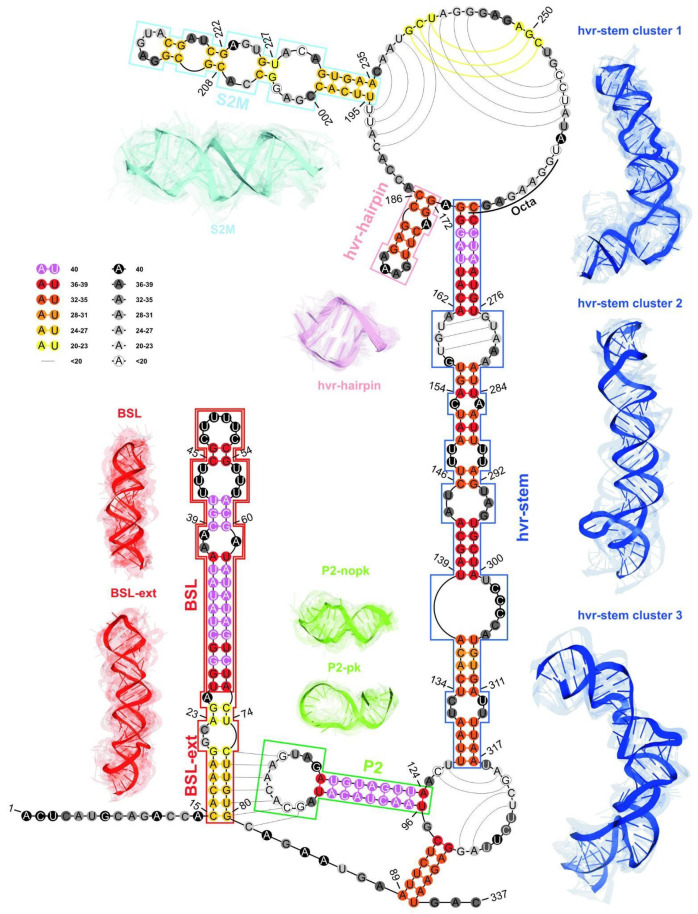
The consensus-driven secondary structure for the 3${}^{\prime }$-UTR region. Domains: BSL (red), P2 (green), HVR hairpin (light purple), S2M (cyan), and HVR stem (blue). Paired regions are coloured according to the preservation of base pairs in all considered 3D RNA models, from magenta (paired in 100% of models) to yellow (paired in ≥50% of models). Unpaired regions are coloured according to the probability that a given residue is not paired in all analysed models, from black (unpaired in 100% of models) to white (unpaired in ≥50% of models). Regions are bordered according to their colouring in 3D models. The centroid of each cluster is solid, and the remaining ensemble members are transparent. The 3D models for the HVR domain are shown for the top three clusters. The P2 domain is shown for two sets of models, pseudoknotted and non-pseudoknotted.

**Figure 9 ijms-23-09630-f009:**
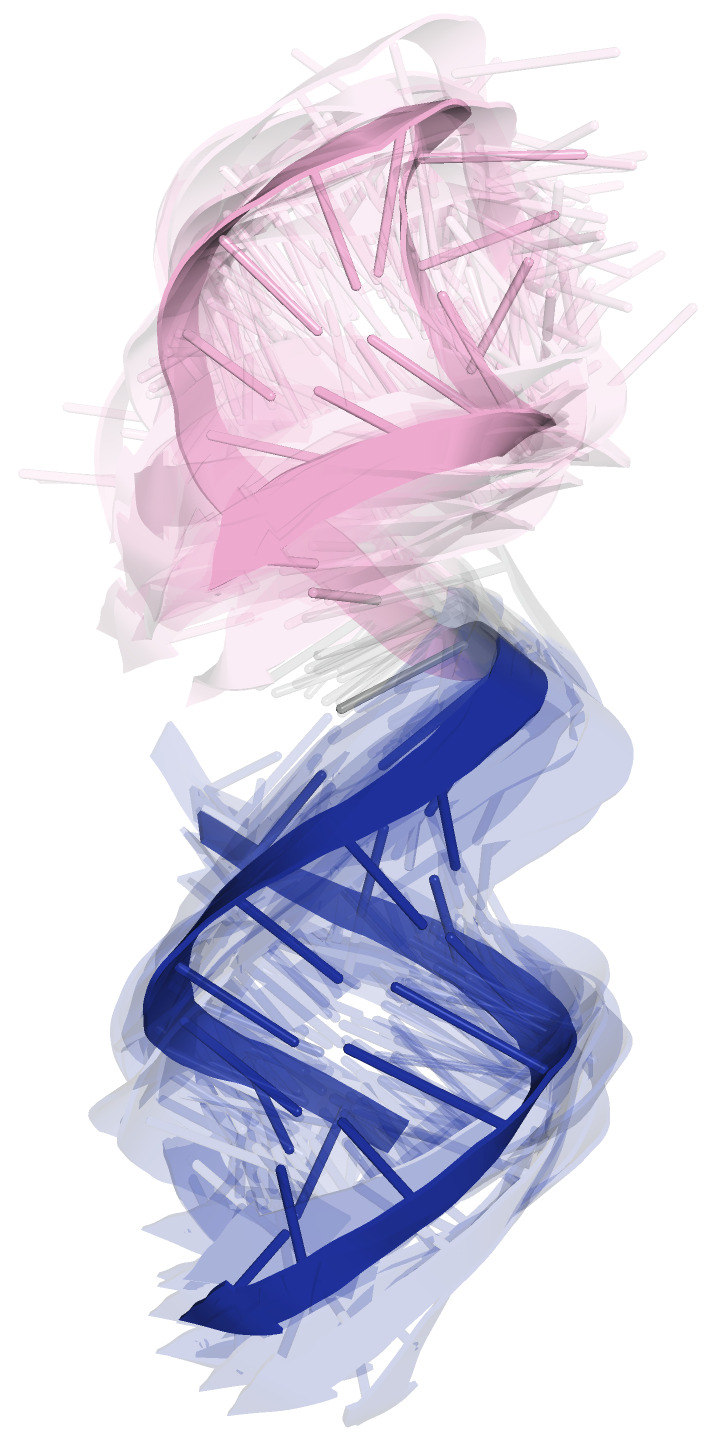
HVR hairpin (light purple) and HVR stem (blue) domains in coaxial arrangement.

**Figure 10 ijms-23-09630-f010:**
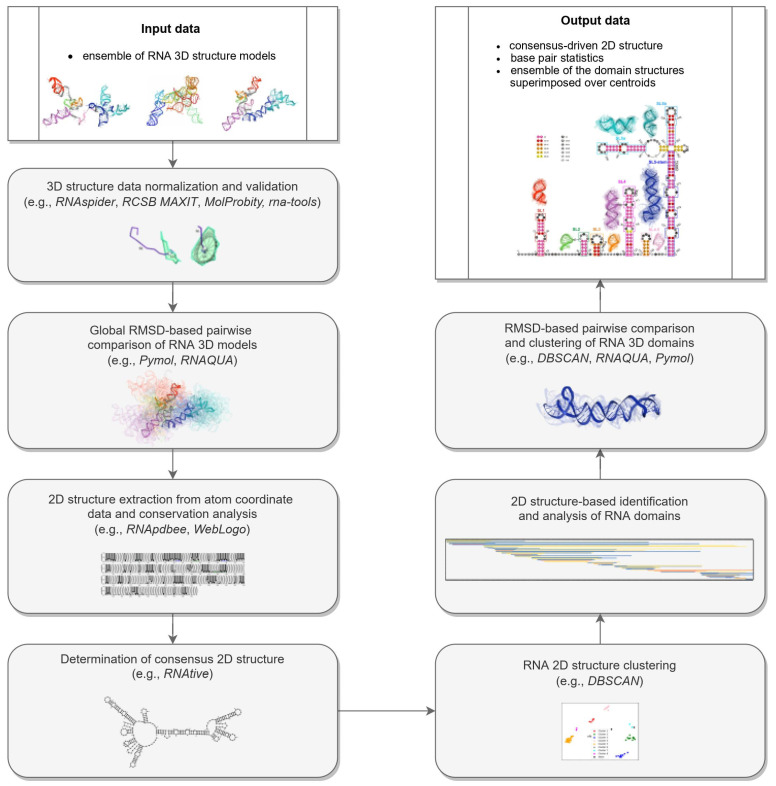
The workflow of the reference-free comparative analysis of RNA 3D structures.

**Table 1 ijms-23-09630-t001:** The number of 3D models predicted within the SARS-CoV-2 RNA-Puzzles challenge.

Modeling	Number of 5${}^{\prime }$-UTR Models			Number of 3${}^{\prime }$-UTR Models			Allf
Group	293–300 nts	450 nts	All	Pseudoknotted *	Non-Pseudoknotted	All	Models
Bujnicki	5	-	5	-	5	5	10
Chen	10	-	10	5	5	10	20
Das	-	10	10	10	-	10	20
Ding	-	-	-	-	10	10	10
Szachniuk	5	4	9	-	5	5	14
Total	20	14	34	15	25	40	74

* Pseudoknot in 3′-UTR models formed between a single-stranded region downstream to BSL and the apical loop of P2.

**Table 2 ijms-23-09630-t002:** The results of the RMSD-based comparison of 3D RNA models for 3${}^{\prime }$ and 5${}^{\prime }$-UTR regions.

Genomic Region	Range (nts)	Length (nts)	# Models	Min–Max RMSD (Å)	Mean RMSD (Å) (Std Dev)	Centroid	Mean RMSD (Å) to Centroid (Std Dev)
5${}^{\prime }$-UTR	1–293	293	34	16.55–64.92	41.05 (7.09)	RNAComposer-5UTRD_4	36.55 (10.26)
3${}^{\prime }$-UTR	10–337	328	40	10.58–84.50	46.51 (9.99)	3UTR-Das-04_1	41.85 (10.00)
3${}^{\prime }$-UTR npk *	1–337	337	25	10.58–84.50	44.81 (12.18)	RNAComposer-3UTRSA1_1	41.45 (16.18)
3${}^{\prime }$-UTR pk *	10–337	328	15	23.73–59.86	39.24 (7.41)	3UTR-Das-07_1	35.25 (11.42)

* npk—nonpseudoknotted, pk—pseudoknotted.

**Table 3 ijms-23-09630-t003:** Results of the RMSD-based domain analysis in 3D models of 5${}^{\prime }$-UTR.

Domain	Range (nts)	Length (nts)	# (%) Models with a Given Domain	Mean RMSD (Å) (Std Dev)	Min–Max RMSD (Å)	Centroid	Mean RMSD (Å) to the Centroid (Std Dev)
SL1	7–33	27	34 (100%)	4.10 (1.31)	0.20–9.05	5UTR-Bujnicki-01	3.32 (1.09)
SL2	45–59	15	34 (100%)	3.07 (0.87)	0.33–4.68	5UTR-Chen-6_8	2.51 (1.14)
SL3	61–75	15	32 (94%)	4.26 (1.25)	0.08–7.69	5UTR-Das-07_1	3.43 (1.38)
SL2+SL3	45–75	31	32 (94%)	8.74 (2.50)	1.07–14.21	RNAComposer-5UTRD_4	7.54 (2.88)
SL4	84–127	44	34 (100%)	5.70 (2.00)	1.65–12.47	5UTR-Bujnicki-04	4.63 (1.56)
SL4a	131–145	15	27 (79%)	3.01 (0.93)	0.29–6.21	5UTR-Chen-6_5	2.56 (0.76)
SL5a	188–218	31	34 (100%)	4.42 (1.40)	0.35–9.24	5UTR-Bujnicki-03	3.66 (1.03)
SL5b	228–252	25	34 (100%)	4.08 (1.05)	0.88–7.01	5UTR-Das-03_1	3.56 (0.95)
SL5 stem	151–182, 263–293	63	34 (100%)	7.77 (2.43)	1.34–15.90	RNAComposer-5UTRE5	6.17 (2.47)
4WJ (0,0,0,0) (4-way junction)	180–185, 225–230, 250–255, 260–265	24	24 (71%)	10.71 (4.67)	0.58–16.56	RNAComposer-5UTRE3	8.90 (6.27)

**Table 4 ijms-23-09630-t004:** The results of the analysis of the RMSD-based domain in the 3D models of 3${}^{\prime }$-UTR.

Domain	Range (nts)	Length (nts)	# (%) Models with a Given Domain	Mean RMSD (Å) (Std Dev)	Min–Max RMSD (Å)	Centroid	Mean RMSD (Å) to the Centroid (Std Dev)
BSL	26–72	47	39 (98%)	6.17 (2.04)	0.04–12.93	3UTR-Das-07_1	4.83 (2.26)
BSL ext	15–80	66	24 (60%)	8.49 (2.54)	2.39–16.54	3UTR-Ding_6	7.11 (3.62)
P2	96–124	29	38 (95%)	6.65 (1.90)	0.04–12.16	RNAComposer-3UTRSA1_5	5.45 (1.79)
P2 npk	96–124	29	23 (92%)	5.53 (1.48)	0.56–10.48	RNAComposer-3UTRSA1_5	4.64 (1.65)
P2 pk	96–124	29	15 (100%)	6.36 (2.20)	0.04–12.16	3UTR-Das-06_1	5.23 (2.31)
HVR hairpin	172–186	15	35 (88%)	2.78 (1.03)	0.03–5.09	3UTR-Chen-2_3	2.20 (1.15)
S2M	195–235	41	24 (60%)	7.82 (2.53)	1.23–13.78	3UTR-Das-01_1	6.42 (2.49)
HVR stem	128–170, 268–317	95	33 (83%)	14.11 (5.16)	0.63–30.55	3UTR-Chen-1_5	11.23 (6.02)
Cluster 1	128–170, 268–317	95	15 (38%)	8.77 (3.56)	0.63–14.63	3UTR-Chen-1_4	6.95 (4.70)
Cluster 2	128–170, 268–317	95	5 (13%)	8.77 (1.82)	6.34–12.11	3UTR-Das-03_1	7.34 (3.81)
Cluster 3	128–170, 268–317	95	5 (13%)	6.12 (2.91)	1.65–11.39	RNAComposer-3UTRSA1_3	5.03 (2.83)

## Data Availability

Any data or material that support the findings of this study can be made available by the corresponding author upon request.

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
