# Peer review of "Computational Pipeline for Reference-Free Comparative Analysis of RNA 3D Structures Applied to SARS-CoV-2 UTR Models"

_ijms, 2022, doi:10.3390/ijms23179630_

Round 1

Reviewer 1 Report

The paper is very interesting. The references are missing and there are [??] throughout the text making it difficult to follow. The authors state 3d models for 5'- and 3'-UTR regions agree with SHAPE or DMS, references missing. I am looking for cross reference against known references such as: Robertson, M.P., Igel, H., Baertsch, R., Haussler, D., Ares Jr., M., Scott, W.G. The structure of a rigorously conserved RNA element within the SARS virus genome 
Plos Biol.3, pp. 86 - 94, 2005.

A comparison between the S2M region in the authors paper and the reference above would be useful and additional such references should be searched out from databases such as NDB and the journal RNA.

A brief discussion on the possible effects modifications such as the effects of m6A methylation in the 5'- and 3'-UTR sequences forming pairs is also warranted, if the authors have any data on the subject

Author Response

We thank the Reviewers for their comments and suggestions. We attempted to answer those points fully. Some of them required modifications in the manuscript, including moderate English changes suggested by the first Reviewer. Major changes have been marked in red.

Referee 1

Comment 1: The paper is very interesting. The references are missing and there are [??] throughout the text making it difficult to follow. The authors state 3d models for 5'- and 3'-UTR regions agree with SHAPE or DMS, references missing. I am looking for cross reference against known references such as: Robertson, M.P., Igel, H., Baertsch, R., Haussler, D., Ares Jr., M., Scott, W.G. The structure of a rigorously conserved RNA element within the SARS virus genome Plos Biol., 3, pp. 86 - 94, 2005.

Response: In fact, due to a compiler error, the manuscript lacked literature references. The problem has been now resolved. The paper indicated by the Reviewer has been cited as item 70 in the References. On the other hand, we disagree with the comment about SHAPE and DMS. Nowhere in the text do we compare the predicted 3D models to data obtained from SHAPE or DMS. When comparing the model to experimental data, we always mean the secondary structure extracted from the 3D model, and we always give references then.

Comment 2: A comparison between the S2M region in the authors paper and the reference above would be useful and additional such references should be searched out from databases such as NDB and the journal RNA.

Response: A comparison to the S2M model from the paper the Reviewer mentions is in the Results section (and contains references). The same is for the SL2 hairpin, for which there are NMR data. We suppose that these sections did not catch the Reviewer's attention. Unfortunately, we are not aware of other papers that provide experimental data on the 3D structure of RNA coronaviruses.  

Comment 3: A brief discussion on the possible effects modifications such as the effects of m6A methylation in the 5'- and 3'-UTR sequences forming pairs is also warranted, if the authors have any data on the subject.

Response: The presence of m6A on RNA can alter local RNA secondary structure exerting stabilizing or destabilizing effects, depending on its sequence context. However it was shown that destabilizing effects are more predominant transcriptome‐wide (Kierzek & Kierzek, 2003; Roost et al, 2015; Spitale et al, 2015). 41 potential modification sites have been identified in the SARS-COV-2 genome (Kim et al, 2020). Among them the most frequently observed motif is AAGAA. It was also shown that m6A modification occur more frequently towards the 3′ end of the SARS-COV-2 genome (Liu et al, 2021; Batista-Roche et al, 2022). However, to our knowledge, all identified modification sites are located in the coding region. Therefore, we do not discuss the subject of modifications in our manuscript.

Reviewer 2 Report

In this work the authors present a computational methodology for  the reference-free high-throughput comparative analysis of RNA  3D structures.

THE WHOLE WORK IS VERY INTERESTING AND COULD HAVE POTENTIAL APPLICATIONS.

POINTS FOR IMPROVEMENT :

1. Please, validate your computational methodology against simplified cases.

2. Please, write a short text explaining how the algorithm converge to the right point. In other words, why the error is not accumulated from the previous steps.

3. Please, provide computational details such as CPU, parameters used, test of parameters used, hardware, etc.

4. Please, expand the reported applications

Author Response

We thank the Reviewers for their comments and suggestions. We attempted to answer those points fully. Some of them required modifications in the manuscript, including moderate English changes suggested by the first Reviewer. Major changes have been marked in red.

Referee 2

In this work the authors present a computational methodology for  the reference-free high-throughput comparative analysis of RNA  3D structures. The whole work is very interesting and could have potential applications. Points for improvement:

Comment 1: Please, validate your computational methodology against simplified cases.

Response: The main goal of the proposed method is the identification of conserved 3D RNA motifs/domains in structures with complex topologies obtained by computational methods to provide clues guiding the process of experimental determination. For "simplified cases" experimentally determined RNA 3D structures are generally known, so the evaluation of related predictions is done based on the reference structure – such procedures are followed in RNA-Puzzles. Therefore, we do not think that a simplified case is a good example. However, we understand the necessity to evaluate the pipeline itself. Fortunately, all the methods incorporated into the pipeline have been already tested in subsequent rounds of RNA-Puzzles, as reported in the RNA-Puzzles papers: Cruz et al., 2012; Miao et al., 2015; Miao et al., 2017; Magnus et al., 2020; Miao et al., 2020.

Comment 2: Please, write a short text explaining how the algorithm converge to the right point. In other words, why the error is not accumulated from the previous steps.

Response: Our pipeline uses a top-down (general to specific) approach. It is the major reason why the errors are not accumulated and propagated to the next steps. We first analyze the global RMSD. Next, we perform comparative analysis and clustering of extracted secondary structures. Then, we extract domains and analyze them separately by computing RMSD values and performing their clustering. Finally, we determine the consensus-driven secondary structure. Every processing step in the pipeline is independent. At each step, we perform a complete computational process, regardless of the results obtained on the other levels. Error accumulation usually occurs when the analysis is conducted in the opposite direction (bottom-up).

Comment 3: Please, provide computational details such as CPU, parameters used, test of parameters used, hardware, etc.

Response: The following information has been now included in the manuscript: Computational experiments presented were run on a mobile workstation equipped with Intel Core i7-10850H CPU @ 2.70GHz, RAM 64.0 GB, Ubuntu 18.04 LTS 64-bit. The consensus-driven evaluation was performed using RNAtive with all base pairs considered and a confidence level equal to 0.5. DBScan was executed with the minimal number of samples in a cluster equal to 2. The maximum distance between two samples for one to be counted as in the neighborhood of the other was 40 and max(10.0, 1.15*AVG RMSD) to cluster non-pseudoknotted secondary structures and domain RMSD scores. Scripts to perform the DBscan clustering and create secondary structure logos were developed in Python using the scikit-learn and weblogo libraries. Scripts to transform the data and process the results were developed in Bash and AWK.

Comment 4: Please, expand the reported applications.

Response: We agree that the presented pipeline can be applied for various problems. We have included the following paragraph in the Conclusions section: The presented pipeline has the potential for various applications. It is useful in comparative analysis of homologs aimed at identifying templates for homology modeling. It can also be applied to classify three-dimensional structures leading to reliable function prediction of biological molecules.

Round 2

Reviewer 1 Report

I have no additional comments

Reviewer 2 Report

I am happy with the revision